



# Retrievals of vertically resolved aerosol microphysical particle parameters with regularization from spaceborne Aerosol and Carbon dioxide Detection Lidar (ACDL)

Ziyu Bi[1,3], Jianbo Hu[1,3], Yuan Xie[1,2], Decang Bi[1,2,3], Xiaopeng Zhu[1,2,3], Jiqiao Liu[1,2,3], Weibiao Chen[1,2,3]

[1]Wangzhijiang Innovation Center for Laser, Aerospace Laser Technology and System Department, Shanghai Institute of Optics and Fine Mechanics, Chinese Academy of Sciences, Shanghai 201800, China
[2]Shanghai Key Laboratory of All Solid-State Laser and Applied Techniques, Shanghai Institute of Optics and Fine Mechanics, Chinese Academy of Sciences, Shanghai 201800, China
[3]Center of Materials Science and Optoelectronics Engineering, University of Chinese Academy of Sciences, Beijing 100049,
China

*Correspondence to*: Jiqiao Liu (liujiqiao@siom.ac.cn), Weibiao Chen (wbchen@siom.ac.cn)

**Abstract.** Using an improved regularization method, we attempt to derive microphysical parameters (effective radius $r_{eff}$, surface area concentration $S_t$, volume concentration $V_t$) of aerosol particle size distribution directly from the detection results of Aerosol and Carbon dioxide Detection Lidar (ACDL), which is the first spaceborne high spectral resolution lidar (HSRL)
based iodine filter. The backscatter and extinction coefficients at 532 nm, 1064 nm, 1572 nm are adopted for regularization algorithm. Preliminary simulations for different aerosol types with monomodal and bimodal lognormal size distributions demonstrate the algorithm performance of the 3α+3β optical data combination. For monomodal aerosols, the retrieval errors are constrained within 15% for $r_{eff}$, 30% for $S_t$, and 35% for $V_t$. In bimodal cases, errors increase to 18-35% for $r_{eff}$, 35% for $S_t$, and up to 60% for $V_t$. Sensitivity analysis confirms that systematic errors of ±20% in input optical data induce
parameter uncertainties below 60%. Case studies reveal four typical aerosols profiles: urban ($r_{eff}$~0.47 μm), smoke ($r_{eff}$~0.57 μm), dust ($r_{eff}$~0.61 μm), and marine ($r_{eff}$~0.83 μm). The inversion $r_{eff}$ is compared with CALIPSO and LIVAS, which confirms high consistency for marine and dust, while urban and smoke retrievals show slightly larger. The inclusion of 1572 nm significantly enhances coarse-mode retrieval accuracy. The error statistics of the simulations and the actual comparison results show that the proposed inversion algorithm can reliably derive the particle size distribution
parameters from the spaceborne multi-wavelength lidar ACDL. This work provides preliminary validation of ACDL's capability to retrieve vertically resolved global aerosol microphysical characterization.

## Introduction

Aerosols are dispersed particle systems consisting of atmospheric medium and the solid or liquid particles suspended in the atmosphere (Mao et al., 2002). Atmospheric aerosols impact the Earth's radiation budget balance through scattering and
absorption of solar radiation, cooling or warming the Earth-atmosphere system (Ipcc, 2007). Furthermore, aerosols act as



cloud condensation nuclei (CCN) and ice-nucleating particles (INPs) and eventually evolve into cloud droplets, which indirectly influence cloud albedo effects. In addition, aerosols-cloud interactions (ACIs) perturb cloud properties, life cycle and precipitation patterns (Patel et al., 2024). It is clear that aerosol particles play a vital role in the Earth's energy balance and climate change. However, aerosols are highly variable and complex both in time and space, which makes monitoring of aerosols arduous. Lack of fully determined and known of the aerosol nature leads aerosol to being one of the largest uncertainties assessments in climate forcing (Ipcc, 2013). The microphysical properties of the aerosol associated with radiative properties are needed to accurately assess the impact of aerosols on climate forcing. Difficulty in achieving the global coverage of aerosol monitoring with traditional ground-based in-situ instruments measurements, while spaceborne multi-wavelength (MW) lidar is capable of providing aerosol observations with a continuous spatial and temporal distribution on a global scale, which is a momentous and fundamental tool for probing the complex physical processes and properties of aerosols. The Aerosol and Carbon dioxide Detection Lidar (ACDL) onboard AEMS (Atmospheric Environment Monitoring Satellite, also named as DQ-1) gives a novel perspective on aerosol profile measurements and scientific research with high accuracy (Chen et al., 2023a).

During the past 30 years, numerous studies have attempted to obtain information on aerosol microphysical properties from MW lidar. Regularization algorithm performed by generalized cross-validation was first tried in the Institute for Tropospheric Research in Leipzig to retrieve aerosol particle size distribution (APSD) from MW Raman lidar (Müller et al., 1999). The algorithm used backscatter coefficients at six wavelengths and extinction coefficients at two wavelengths, which was possible to give reliable results of aerosol microphysical parameters (AMP). Veselovskii et al. used Tikhonov's inversion with regularization to derive AMP from Mie-Raman lidar, and extended the particle size range to 10 μm, which made the algorithm applicable to large particles (Veselovskii et al., 2004). Besides, averaging the solutions was performed to further stabilize the inversion results (Veselovskii et al., 2002). In the case of known refractive index distribution, principal component analysis method was chosen to retrieve aerosol bulk physical and optical properties from MW Raman lidar (Donovan and Carswell, 1997). Li et al. attempted to reduce the aerosol optical data required for the algorithm, using 3-4 optical parameters for inversion based on the principal component analysis (Li et al., 2023). Linear estimation algorithm was used to estimate aerosol concentration directly from characteristics measured by MW lidar without deriving the APSD (Veselovskii et al., 2012).

Among the several algorithms mentioned above, the regularization algorithm requires neither a priori estimation of the error in the original data nor a priori assumptions about the systematic error. Thus, the algorithm is widely used and further developed by other studies. Kolgotin et al. retrieved microphysical properties of pollution aerosol with regularization algorithm. The algorithm used three backscatter coefficients and two extinction coefficients obtained by NASA Langley Research Center's airborne high-spectral-Resolution lidar (HSRL-2), and found good agreement between inversion results and in situ measurements (Kolgotin et al., 2023). Significant progress was also made within the framework of the NASA Aerosol-Clouds-Ecosystems (ACE) mission, which served as the baseline for the upcoming Atmospheric Observing System (AOS) mission. Whiteman et al. (Whiteman et al., 2018) used HSRL-2 simulations to assess the capability of inversion by





regularization for aerosol microphysical retrievals and highlighted the limitations. Subsequently, Tikhonov Advanced
    Regularization Algorithm (TiARA) was proposed, which is an automated and unsupervised algorithm for the autonomous
    retrieval of AMP. The credibility of TiARA was confirmed by extensive simulation studies (Müller et al., 2019). In parallel,
    case-dependent optimized-constraints were proposed to further optimize regularization. Perez-Ramirez et al. demonstrated
    improved retrievals of aerosol single scattering albedo and other AMP with HSRL-2 measurements (Pérez-Ramírez et al.,

2019), and then extended this approach to simulated spaceborne lidar observations and confirmed its potential for future
    satellite missions (Pérez-Ramírez et al., 2020). Wang et al. developed a machine learning method for inverting MW lidar
    data into AMP and improved the retrieval of fine mode aerosol particle parameters (Wang et al., 2022). Moreover, to address
    particle shape effects, Veselovskii et al. applied non-spherical kernel function for dust particles, extending the applicability
    of regularization methods to more realistic aerosol scenarios (Veselovskii et al., 2010).

In this paper, the regularization algorithm is adopted for the first attempt to directly derive AMP from backscatter signals
    observed with spaceborne lidar ACDL. The algorithm inputs optical data obtained by ACDL consist of vertical profiles of
    three backscatter coefficients and three extinction coefficients measured at 532 nm, 1064 nm and 1572 nm, and outputs the
    AMP (effective radius, surface area concentration and volume concentration). Based on the use of simulation models to
    verify the reliability of the algorithm, this work focuses on the inversion results of microphysical properties of four typical

aerosols detected by ACDL.

    The paper is organized as follows. The introduction addresses the significance for research on APSD and AMP. Section 2
    shows the data sources for the inversion of aerosols and modelling, in particular the satellite observations from ACDL.
    Section 3 describes the algorithmic principles and flow. Section 4 encompasses simulation inversion, extensive validation
    efforts and sensitivity analysis. Case studies of observations of four typical aerosols with ACDL are discussed in section 5.

Finally, section 6 gives a summary.

## 2 Dataset

### 2.1 ACDL

    AEMS was launched successfully on April 16, 2022, which is a new-generation satellite including active and passive
    instruments for environment monitoring and climate changing. The satellite provides global comprehensive measurement of

carbon dioxide ($CO_2$), atmospheric clouds and aerosols particles, and has been in continuous operation for more than three
    years in a sun-synchronous orbit at the altitude of 705 km. The ACDL, as the primary payload onboard AEMS, integrates
    two lidar sub-systems into one lidar system and simultaneously and outputs single-beam with double-pulse lasers at three
    wavelengths (532 nm, 1064 nm and 1572 nm) (Chen et al., 2023b). One is the high-spectral-resolution lidar (HSRL) (Dong
    et al., 2018), which gives vertical profiles measurements of clouds and aerosols with high accuracy (Liu et al., 2024). The

other is an integrated path differential absorption (IPDA) lidar (Zhu et al., 2021), which provides all-time and high-accuracy
    observations for atmosphere column $CO_2$ (Fan et al., 2024). In previous studies, the inversion algorithms for processing the



echo backscatter signals collected by the ACDL (Dai et al., 2024) and further obtaining high-precision aerosol type classification profiles (Hu et al., 2023) have been described in detail. This paper provides insight into the ability of ACDL to derive microphysical parameters for the four typical aerosol types using aerosol extinction coefficient profiles, backscattering coefficient profiles, and aerosol classification results from data products measured with ACDL.

## 2.2 AERONET

The Aerosol Robotic Network (AERONET) is a ground-based aerosol remote sensing network consisting of globally distributed sun photometers, which are capable of automatic tracking and scanning. Generally, the aerosol optical depth (AOD) obtained by AERONET are used as truth values to assess the accuracy of remote sensing measurements. In addition, long-term continuous and easily accessible data on the optical, microphysical and radiative properties of aerosol are provided for optimization of inversion algorithms, data assimilation and aerosol monitoring research (Giles et al., 2019). Based on many years of AERONET radiometer observations, Dubovik et al. have given the optical characteristics for desert dust, biomass burning, urban-industrial, and marine aerosols (Dubovik et al., 2002). In this paper, the summary of aerosol optical properties is used in simulation to randomly generate different types (urban aerosol, dust and marine, biomass burning) of lognormal aerosol models.

## 2.3 CALIPSO and LIVAS

Cloud Aerosol Lidar and Infrared Pathfinder Satellite Observation (CALIPSO) launched successfully on April 28, 2006, which aims to observe the vertical structure and properties of clouds and atmospheric aerosols (Hunt et al., 2009). Cloud-Aerosol Lidar with Orthogonal Polarization (CALIOP) is the primary payload of the CALIPSO, which emits pulsed laser at two wavelengths (532 nm and 1064 nm) simultaneously and is operated in 705 km sun-synchronous polar orbit until the end of the mission in August 2023 (Winker et al., 2009). CALIPSO has proven data processing solutions and the ability to identify and characterize a wide range of aerosols (Kim et al., 2018). The CALIPSO aerosol models are developed based on cluster analysis of the AERONET multiyear dataset (Omar et al., 2005), using almost simultaneous observations of physical and optical properties to determine the type of aerosol (Omar et al., 2009). The model contains six representative aerosol types, which are biomass burning, polluted dust, polluted continental, clean continental, marine and dust.

Lidar climatology of Vertical Aerosol Structures (LIVAS) is a 3-D MW global aerosol and cloud optical database proposed by the National Observatory of Athens (NOA), the Leibniz Institute for Tropospheric Research (TROPOS) in Germany, and the Institute of Methodologies for Environmental Analysis in Italy. Based on CALIPSO observations and ground-based EARLINET measurements, the LIVAS provides the average profiles of aerosol optical properties at five operating wavelengths (355 nm, 532 nm, 1064 nm, 1570 nm and 2050 nm), and the average profile of optical properties of clouds at 532 nm, providing a realistic and reliable environment simulation for satellite observation missions (Amiridis et al., 2015). The LIVAS aerosol model consists of six typical aerosol types, namely dust, polluted dust, clean continental, polluted continental, smoke and marine.





Four aerosol types (urban, smoke, dust and marine) of the representative aerosol classification provided by CALIPSO model
and LIVAS model are used in this paper. Subsequently, based on the APSD, the microphysical parameter, i.e., the effective
radius is calculated to compare the data with the ACDL inversion results and assess the credibility of the ACDL inversion
results.

## 3 Methodology

Most of the previous studies used the backscattering coefficients at 355 nm, 532 nm, 1064 nm and the extinction coefficients
at 355 nm, 532 nm to invert the APSD and AMP, which is also the most classical combination of the input optical data.
However, considering the operating wavelengths of the ACDL, both the backscattering coefficients and the extinction
coefficients at 532 nm, 1064 nm and 1572 nm are chosen for the regularization algorithm.

The optical data obtained from lidar are related to the microphysical properties by Fredholm integral equations of the first
kind in the following form:

$$g_i(\lambda) = \int_{r_{min}}^{r_{max}} K_i(r,m,\lambda)v(r)dr, \tag{1}$$

where $g_i(\lambda)$ is the optical data at the different operating wavelengths $\lambda$, $i$ denotes aerosol backscattering coefficient ($\beta$) and
extinction coefficient ($\alpha$). $r$ is the particles radius. The upper limit of integration of Eq. (1) is determined by $r_{max}$ , and the
lower limit of the integration is determined by $r_{min}$. $v(r)$ describes the volume concentration of aerosol particles per radius
interval $dr$. $K_i(r,m,\lambda)$ represents the volume kernel function for backscatter or extinction, which depend on the particle
radius $r$, the complex refractive index $m$, and the incident wavelength $\lambda$. In case the particles are assumed to be spherical, the
kernel function can be calculated from the Mie theory.

Further information about the aerosol effective radius, volume concentration and surface area concentration can be derived
from the volume concentration distribution. Nevertheless, the volume cannot be solved analytically from the optical data,
which will lead to ill-posed problem. Therefore, regularization algorithm is proposed here. The main processes are described
below.

The solution of the Eq. (1) can be approximated by a linear superposition of base functions as follows:

$$v(r) = \sum_j \omega_j B_j + \varepsilon, \tag{2}$$

where $\omega_j$ are weight factors. $B_j$ are base functions and B-spline functions of first degree are chosen here. $\varepsilon$ represents the
mathematical error due to approximation. The subscript $j$ marks the different base functions and weight factors, which is
numerically equal to the total number of optical data. According to Eq. (1) and Eq. (2), the optical data can be written as:

$$g_i = \sum_j A_{ij}(m)\omega_j + \varepsilon, \tag{3}$$

where kernel function matrix $A_{ij}$ is calculated from the respective kernel function and the base function:





$$A_{ij} = \int_{r_{min}}^{r_{max}} K_i(r,m) B_j(r) dr. \tag{4}$$

By representing the optical data $g$, the weighting factors $\omega$ as vectors, the weight factor matrix can be derived as:

$$\boldsymbol{\omega} = (\mathbf{A}^{\mathrm{T}}\mathbf{A} + \gamma\mathbf{H})^{-1}\mathbf{A}^{\mathrm{T}}\boldsymbol{g}, \tag{5}$$

where $\mathbf{A}$ is weight matrix, $\mathbf{A}^T$ is the transposed matrix of weight matrix. $\gamma$ denotes Lagrange multiplier, which determined by generalized cross-validation. $\mathbf{H}$ represents the smoothness matrix. Substituting the solution of Eq. (5) into Eq. (2) and then linearly superimposing it with the base functions allows ultimate reconstruction of the aerosol volume concentration distribution.

Both the complex refractive index and the particle radius of aerosols have a significant effect on optical properties. Thus, a lookup table (LUT) is created in simulation to try all possible complex refractive index (for real parts of the refractive index, values of 1.3-1.6 are used, and for the imaginary part, values of 0.001-0.2 are chosen) reconstruct the volume concentration distribution with a gliding inversion window. This process leads to extensive inversion results, however not all of them are expected and acceptable. Results containing negative values are first excluded, due to the non-negative nature of the volume

concentration distribution. To further improve inversion accuracy, a criterion is introduced as the following equation:

$$\rho = \frac{1}{M}\sum_p \frac{\left\| g_p - K_p v(r) \right\|}{g_p}, \tag{6}$$

where $M$ is numerically equal to the total amount of optical data. The subscript $p$ marks the input optical data and the kernel functions. Optical data are derived again from the volumetric concentration distribution obtained by inversion, which is needed to calculate the average relative error (ARE) with the optical data from lidar measurements. An average of a large

number solutions near the minimum ARE should be regarded as the final solution. The average process avoids under fitting problem associated with a particular individual solution, which is expressed as follows:

$$v_{mean} = \frac{1}{N}\sum_{k=1}^{N} v_k(r), \tag{7}$$

where $N$ denotes the number of solutions, and $v_{mean}$ indicates the final volume concentration distribution after averaging. The effective radius, surface area concentration and volume concentration of the aerosol can be further revealed from $v_{mean}$

which is calculated as follows:

$$r_{eff} = \frac{\int v_{mean}(r) dr}{\int \frac{v_{mean}(r)}{r} dr}, \tag{8}$$

$$S_t = 3\int \frac{v_{mean}(r)}{r} dr, \tag{9}$$

$$V_t = \int v_{mean}(r) dr. \tag{10}$$



## 4 Simulation

### 4.1 Lognormal aerosol size distribution

As the acceleration of industrialization pace, APSD becomes more complex and diverse. Numerous studies have shown that the lognormal distribution, especially the bimodal lognormal distribution, better describes the aerosol scale distribution. The lognormal distribution can be expressed as:

$$\frac{dv(r)}{dlnr} = \sum_{\delta} \frac{C_{V,\delta}}{\sqrt{2\pi}\sigma_\delta} exp\left[-\frac{\left(lnr - lnr_{V,\delta}\right)^2}{2\sigma_\delta}\right], \tag{11}$$

where subscript $\delta$ indicates the number of aerosol modes. $C_{V,\delta}$ stands for the particle volume concentration of the $\delta$ mode (fine mode and coarse mode). $\sigma_\delta$ is the standard deviation, and $r_{V,\delta}$ is the median radius. Aerosol simulation is developed in order to verify the feasibility and assess the accuracy of the algorithm. Aerosols are generally classified into four typical categories (urban industrial aerosol, biomass burning aerosol, desert dust blown into the atmosphere, and marine aerosol) associated with different sources and emission mechanisms. In this paper, the marine aerosols are simulated together with dust aerosols, both dominated by coarse modes. Assuming the particles are spherical, aerosol models are defined as urban aerosol, dust and marine, biomass burning with unimodal and bimodal lognormal distribution. The simulation parameters are shown in Table 1:

**Table 1. Parameters of lognormal distribution for urban aerosol, dust and marine, and biomass burning**

| Aerosol Type | $r_{V,f}$ (μm) | $r_{V,c}$ (μm) | $\sigma_f$ | $\sigma_c$ | $C_{V,f}/C_{V,c}$ |
|---|---|---|---|---|---|
| Urban aerosol | 0.14-0.18 | 2.70-3.20 | 0.38-0.46 | 0.60-0.80 | 0.80-2.00 |
| Dust and marine | 0.12-0.16 | 1.90-2.70 | 0.40-0.53 | 0.60-0.70 | 0.10-0.50 |
| Biomass burning | 0.13-0.16 | 3.20-3.70 | 0.40-0.47 | 0.70-0.80 | 1.30-2.50 |

Through the regularization process described in the previous section, the APSD can be reconstructed as Fig. 1. The green solid line represents the given distribution based on the Table 1 and the tangerine dash line represents the retrieved distribution.



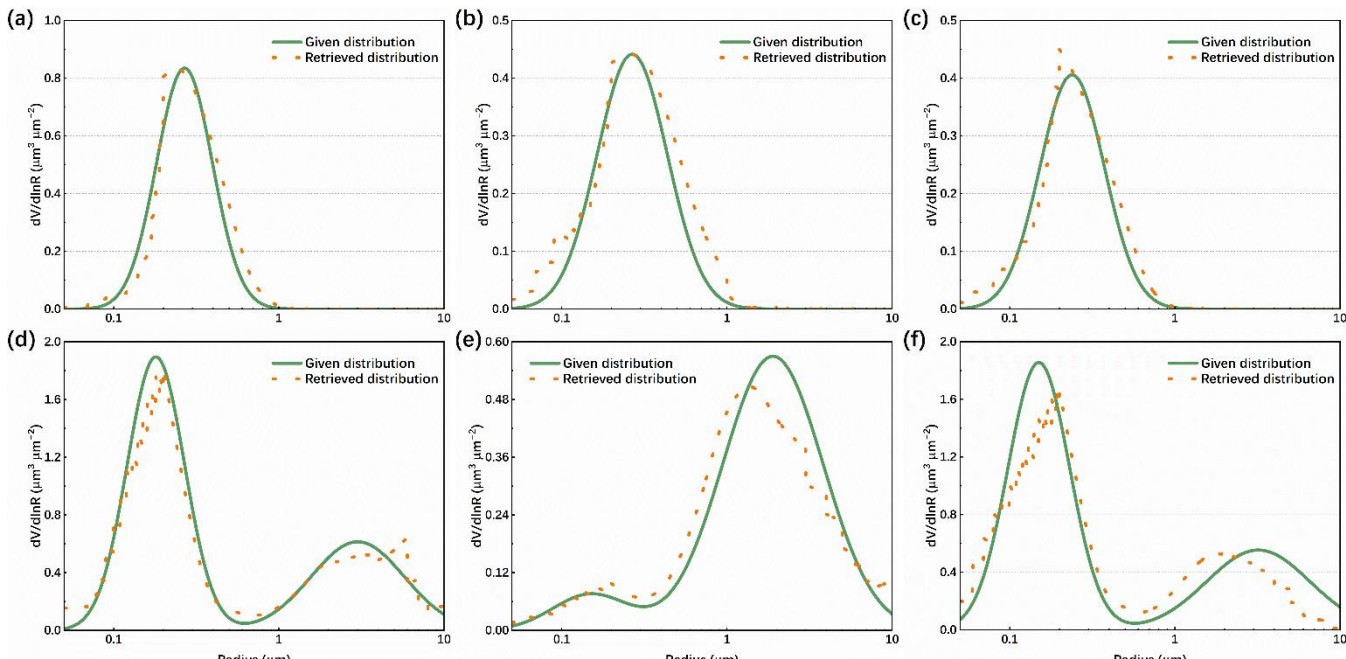

**Figure 1: Retrieved APSD for urban aerosol (a, d), dust and marine (b, e), and biomass burning (c, f).**

As can be seen in Fig. 1, the algorithm is capable to reconstruct the volume concentration distribution of the aerosol with acceptable results. The inversion APSD has good consistency with the given unimodal lognormal distribution. For bimodal lognormal distribution aerosol models, oscillation occurs at the edge of the inversion radius interval (d), which is due to insufficient optical data input. Furthermore, the inversion results of the APSD are more stable for dust and marine (e), where coarse mode particles are the dominant form, than for urban aerosol (d) and biomass burning (f) dominated by fine mode.

The enhancement can be attributed to the introduction of 1572 nm into the wavelength combination selected for regularization, which has a peak extinction efficiency that corresponds to an aerosol particle sensitivity radius of approximately 1-2 µm. This facilitates the acquisition of more microphysical properties of aerosols dominated by coarse mode and is advantageous in the APSD inversion.

## 4.2 AMP inversion

Based on the Eq. (8)-(10), the AMP such as effective radius, surface area concentration, and volume concentration can be derived from the retrieved distribution and given distribution, respectively. The relative error between actual and inverse AMP can now be calculated. A substantial number of models are generated at random in accordance with the parameters set forth in Table 1. Figure 2 illustrates the degree of discrepancy between the initial and retrieved values for the AMP results across the three aerosol models. Each model comprises 10 unimodal distributions and 14 sets of bimodal distributions.





**Figure 2: AMP error statistics for simulations of three aerosol modals. The statistical data for the unimodal distribution are presented on the left side, and the bimodal distribution are shown on the right.**

The inversion errors for effective radius (a, b), surface area concentration (c, d), and volume concentration (e, f) are shown in Fig. 2. The effective radius inversion error for unimodal distribution aerosols is the lowest (a), basically staying below 15%.





The effective radius of the bimodal distribution modals inversion is characterised by an error below 18% for urban aerosol, essentially below ~30% for dusty and marine, and no more than 35% for biomass burning with bimodal distribution (b). It can be seen that the inversion errors for the unimodal distributions are, in general, lower than those for the bimodal distributions. This same pattern is demonstrated in the error statistics for the surface-area concentration and the volume concentration. The maximum permissible errors for surface area concentration inversion are 30% for unimodal aerosols (c)

and are basically below 35% for bimodal distributions (d). The errors in the derived volume concentration for aerosols with unimodal distribution do not exceed 35% (e), while for bimodal distributions are found to be basically below 40%, and up to 60% (f). Analysing the algorithm in terms of its mathematical principles, the process of regularization employs a finite number of discrete base functions, each carrying a distinct weighting factor, to approximate the original distribution curve through superposition. As illustrated in Fig. 1, the unimodal lognormal distribution of aerosols exhibits reduced number of

peaks and more concentrated features in comparison to the bimodal lognormal distribution. In the case of an equivalent number of base functions, the reconstruction results will be more closely aligned with the original APSD curve, and the derived particle size distribution parameters will exhibit a reduced level of error.

### 4.3 Sensitivity analysis

Spaceborne lidar is a complex detection system, and two main unavoidable uncertainties are inherent to its detection process.

The first is the random error caused by the generation of scattering noise, thermal noise, and other forms of noise during the operation of photodetectors. Secondly, defective optics, non-linearities in photodetectors and inaccurate calibration may result in the occurrence of systematic errors in detection. Furthermore, the optical data obtained from the spaceborne lidar signal, which serves as the input to the algorithm, is also susceptible to inversion errors due to factors such as the selected atmospheric mode and lidar ratio (Liu et al., 2006). Notably, ACDL's high-spectral-resolution allows direct retrieval of the

lidar ratio at 532 nm, reducing dependence on atmospheric assumptions. In order to evaluate the sensitivity of the AMP inversion results to errors in the optical data, systematic errors ranging from -20% to 20% are introduced separately into the simulations for the optical data. This range of error is reasonable for the majority of lidar systems. Assuming the systematic errors between individual optical data are independent of one another. The variation of AMP with optical data errors for three typical aerosols are presented in Fig. 3.

Systematic errors of ±20%, ±15%, ±10, ±5% are introduced for the six input optical data respectively. The relative error distributions of effective radius, surface area concentration, and volume concentration for the three types of aerosols are shown in Fig. 3. As the error in the input optical data increases, so does the error in the inverted particle size distribution parameters, but the relationship between the two is not always linear. In general, when systematic errors of between -20% and 20% are introduced to the optical data, the majority of the output inversion results can be controlled within 50% of the

error, up to 60% (b). Of the six input optical data, the inversion results are most sensitive to changes in the 532 nm extinction coefficient, represented as yellow dots in Fig. 3. Previous study has carried out extensive analyses of the impact of systematic and random errors on inversion by regularization, with particular emphasis on the importance of extinction





measurement errors (Pérez-Ramírez et al., 2013). Our sensitivity analysis, based on simulations, is consistent with the findings, showing that when introducing systematic errors of the same magnitude, perturbations in the 532-nm extinction coefficient generally lead to larger retrieval errors. It should be noted that these results are still derived from idealized simulations, however, our goal is to extend the retrievals to real observations from spaceborne lidar system. As the first spaceborne HSRL, ACDL can directly derive extinction coefficients without assuming a lidar ratio, thereby reducing potential biases and is expected to enhance the reliability of AMP retrievals under the real observational conditions.



Figure 3: Relative error of effective radius (a, d, g), surface area concentration (b, e, h), and volume concentration (c, f, i) of three typical aerosols varies with optical data introduced to different systematic errors





# 5 Results

## 5.1 ACDL inversion of AMP

ACDL provides high-precision vertical profiles of cloud and aerosol optical properties (backscatter and extinction
coefficients) at three wavelengths (532 nm, 1064 nm, and 1572 nm), as shown in Figure 4, along with cloud and aerosol
classification results presented in Figure 5. This section presents four aerosol scenarios, extracted from ACDL data products,
which are urban, smoke, dust and marine, respectively. The regularization algorithm described is employed to invert AMP
profile for four typical aerosols. The specific case studies are as follows.

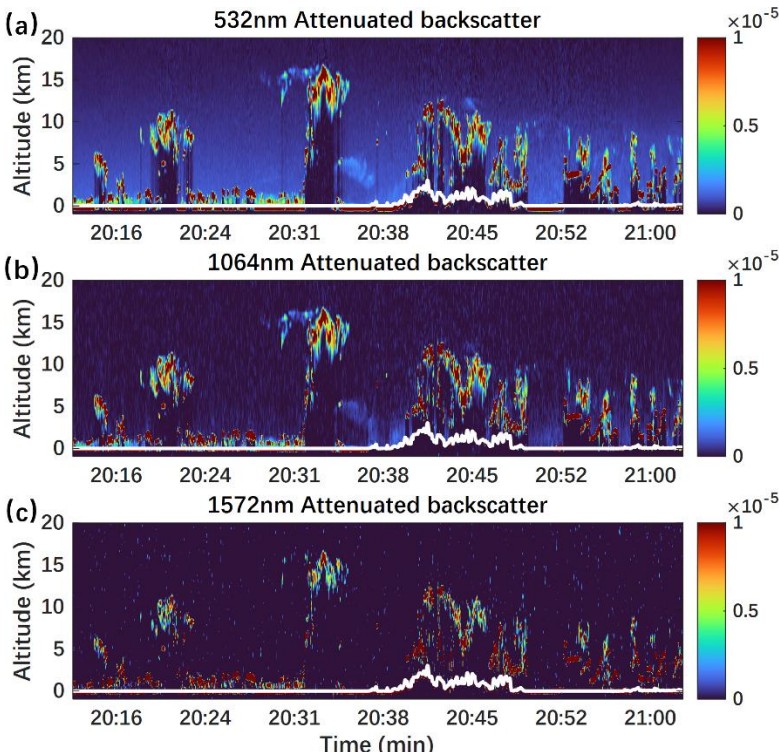

**Figure 4: Attenuated backscatter on 13 June 2023, 20:13-21:02, retrieved from ACDL at 532 nm(a), 1064 nm(b), and 1572 nm(c),**
**respectively. The white line denotes the surface elevation**




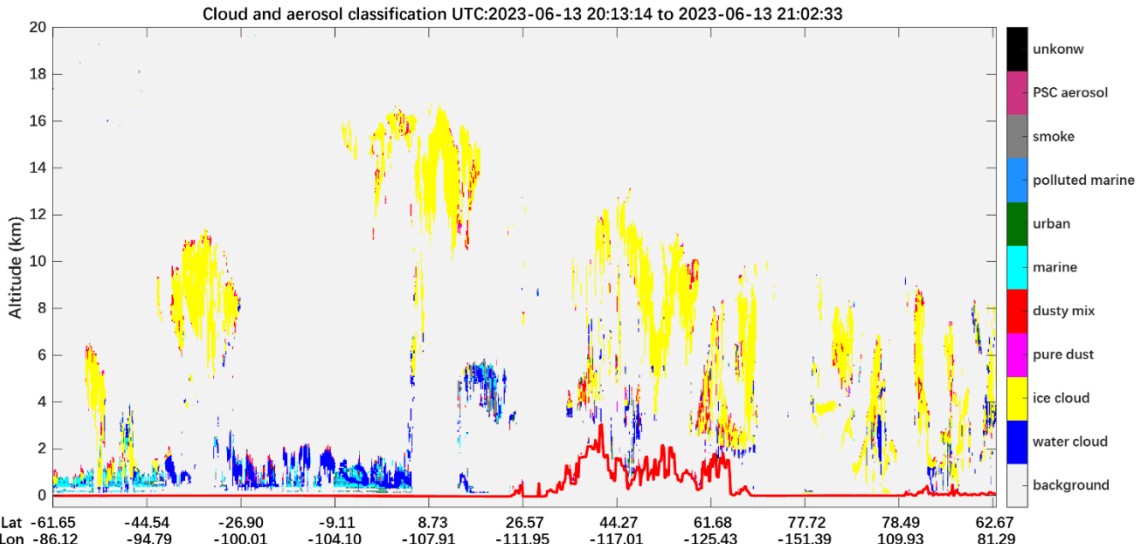

**Figure 5: Cloud and aerosol classification on 13 June 2023, 20:13-21:02, retrieved from ACDL. The red line denotes the surface elevation**


### 5.1.1 Urban aerosol

The formation and evolution of urban aerosols can be influenced by a range of natural and anthropogenic factors, including water vapour and humidity, different emission and input sources, and combustion of various chemical fuels. These factors can lead to notable variations in the optical properties of urban aerosols. One data set recorded at 17:54 Universal Time

Coordinated (UTC) time on the night of 1 March 2023, is selected for analysis. Figure 6 illustrates the urban AMP variation trend obtained from the inversion. The effective radius size of urban aerosols fluctuates around 0.5 μm, as shown in Fig. 6a. Urban aerosol effective radius for this profile range from a minimum of 0.298 μm to a maximum of 0.683 μm. The vertical distribution of surface area concentration and volume concentration are shown in Fig. 6b and 6c, respectively. Both exhibit lower concentrations at the top of this aerosol but a sudden increase in surface area concentration and volume concentration

at the bottom, 5.19 km-5.23 km, and a simultaneous reduction in the effective radius is observed. The particle size of the urban aerosol cluster demonstrates a notable variation across vertical distances, exhibiting a lower concentration distribution at the top and a higher concentration at the bottom, which is predominantly composed of fine mode particles.




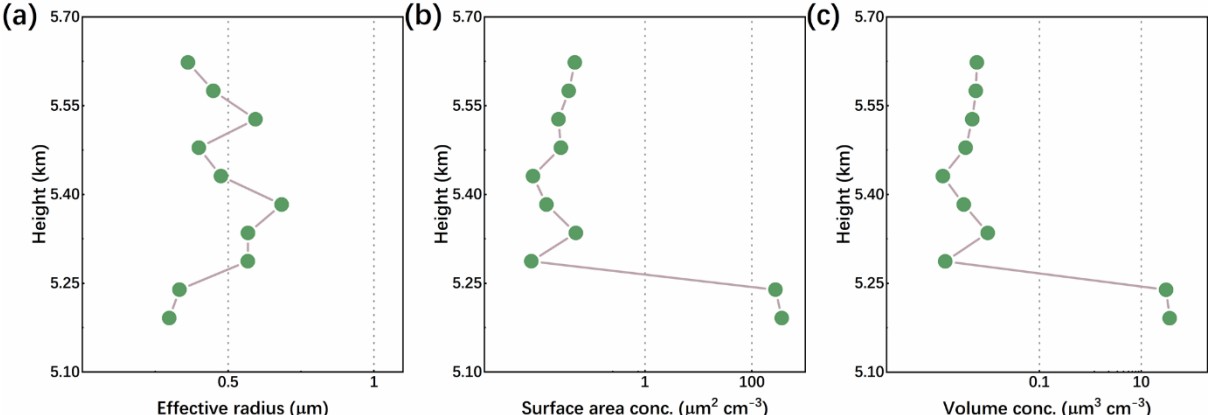

**Figure 6: Urban aerosol AMP inversion from ACDL data products**

### 5.1.2 Smoke aerosol

Smoke aerosol is produced by the biomass burning. The characteristics of smoke aerosol, including particle size, can be influenced by various factors such as the type and quantity of combustible materials, the stage and intensity of combustion, and the environmental conditions, including humidity and temperature (Ansmann et al., 2021). In the summer of 2023, a wildfire event occurred in Canada that lasted for a period exceeding two months. During this time, there were more than 200 monitored fires, spanning an area of 80000 square kilometres. The wildfire produces substantial quantities of smoke, which can have a detrimental impact on coastal regions of the United States and Europe with the atmospheric circulation patterns (Jain et al., 2024). Figure 7 shows the trend of smoke AMP obtained from the inversion of a track of ACDL data on 14 June,2023. As can be seen from the height, this smoke aerosol is distributed in the range of 10.04-10.37 km and has reached the tropopause in altitude. This smoke has a relatively uniform particle size, with an effective radius of around 0.6 μm, with a minimum of 0.42 μm and a maximum of 0.64 μm (a). Surface area concentrations are mainly maintained in the range of 4-8 $\mu m^2 \, cm^{-3}$ (b), and volume concentrations are basically distributed in the range of 0.7-1.8 $\mu m^3 \, cm^{-3}$ (c). Nevertheless, both values are markedly lower at 10.04 km and 10.37 km. The smoke has a more homogeneous particle size distribution and a lower concentration at high altitude. Furthermore, the concentration of smoke particles is significantly lower at the boundary.





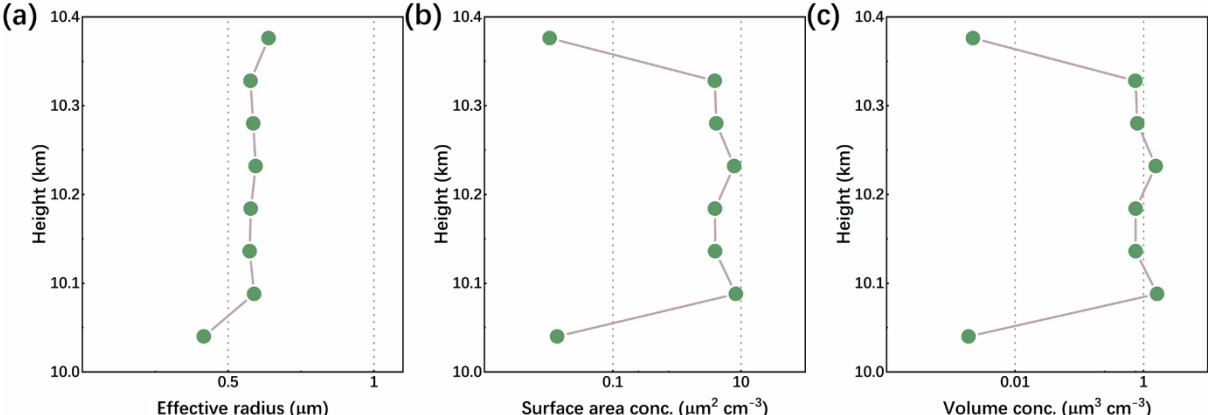

**Figure 7: Smoke aerosol AMP inversion from ACDL data products**

### 5.1.3 Dust aerosol

Atmospheric dust plays a complex role in the climate system and contributes to the deterioration of air quality. The vertical distribution of the particle size distribution is closely related to the life cycle of the transport and settlement of dust particles. Coarse particles typically settle in close proximity to the source, whereas fine particles tend to be transported over longer distances. Coarse dust particles are more likely to converted into CCN or INP, which have an impact on the Earth's marine and terrestrial ecosystems. In contrast, fine mode dust particles are more likely to have an adverse effect on human health (Proestakis et al., 2024). In the spring of 2023, Beijing was subjected to a series of dusty weather events, the formation of which was caused by the high-altitude transport of dust from the deserts and the Gobi of Mongolia, which settled in areas surrounding Beijing. Inversion and analysis of the vertical distribution of dust AMP for the one-track data transiting Beijing on 1 March at UTC time was selected. In order to invert and analyse the vertical distribution of dust AMP, a set of data from the transit of Beijing on 1 March is selected. The results of the dust aerosol profile inversion at low altitude from 0.67 km to 1.05 km are shown in Fig. 8. The effective radius of the dust is predominantly in the range of 0.6-0.7 μm, decreasing abruptly to 0.45 μm at 1km (a). Surface area concentration distribution between 400-1000 $\mu m^2$ $cm^{-3}$ (b). Volume concentrations are mainly distributed between 100-200 $\mu m^3$ $cm^{-3}$, with the lowest at the bottom (0.67km) and it is only 80.32 $\mu m^3$ $cm^{-3}$ (c). It can be surmised that this low-altitude aerosol is predominantly constituted by coarse-modelled dust particles, with a significant increase in the concentration of aerosol clusters at an altitude of 0.7-0.9 km and a slight decrease in the size of the particles. Larger dust aerosols settle at the bottom of the atmosphere and have a relatively low concentration.



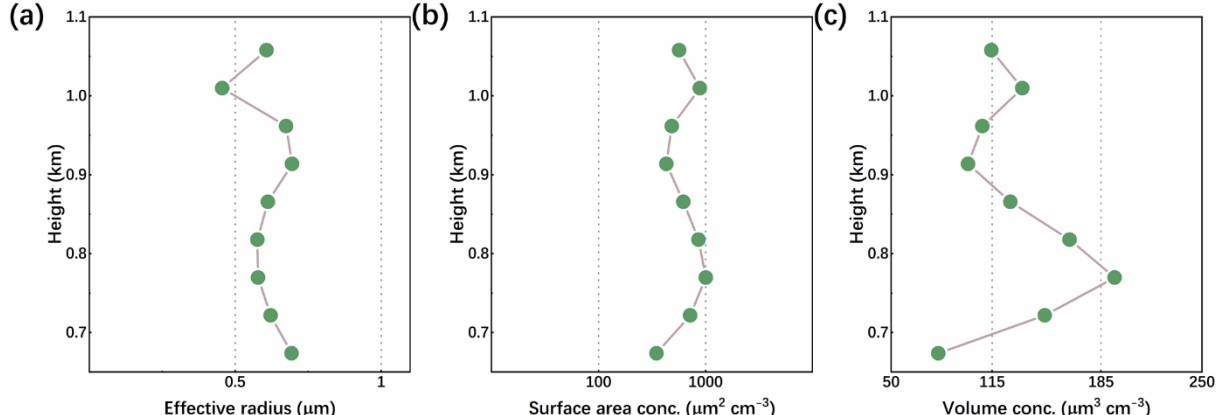

**Figure 8: Dust aerosol AMP inversion from ACDL data products**

### 5.1.4 Marine aerosol

Marine aerosols are formed by sea fog, which carries sea salt and organic matter from the surface of the sea. This process occurs in more than 70 % of the Earth's oceans. Marine aerosols act as CCN and INP over the oceans and are of a relatively homogeneous origin. Consequently, their properties are not influenced by the source of emissions but rather by relative humidity (Haarig et al., 2017). Data from 13 June 2023 are selected for inversion, and the results of the vertical distribution of marine AMP are shown in Fig. 9. The effective radius size of this marine aerosol is mainly distributed within the range of 0.8-0.9 µm (a), which is more evenly distributed over the vertical distance, with a slight increase in the effective radius at 3.7-3.8 km, while at the bottom boundary, the effective radius decreases from 0.95 µm to 0.64 µm. Surface-area concentrations are predominantly distributed between 370-500 $\mu m^2$ $cm^{-3}$, reaching a maximum of 517 $\mu m^2$ $cm^{-3}$, while at the bottom they decrease sharply to below 30 $\mu m^2$ $cm^{-3}$ (b). The volume concentrations are mainly distributed between 100 $\mu m^3$ $cm^{-3}$ and 130 $\mu m^3$ $cm^{-3}$, attaining a maximum of 140 $\mu m^3$ $cm^{-3}$, and subsequently declining to a value below 9.6 $\mu m^3$ $cm^{-3}$ at the lowest height (c). In general, the particle size and concentration of marine aerosol clusters at this site are uniformly distributed over vertical altitude. As the altitude decreases, the marine aerosol particle size decreases significantly at the bottom boundary after becoming slightly larger. Following a uniform distribution over a distance, the concentration of particles declines gradually at the base of the marine aerosol.





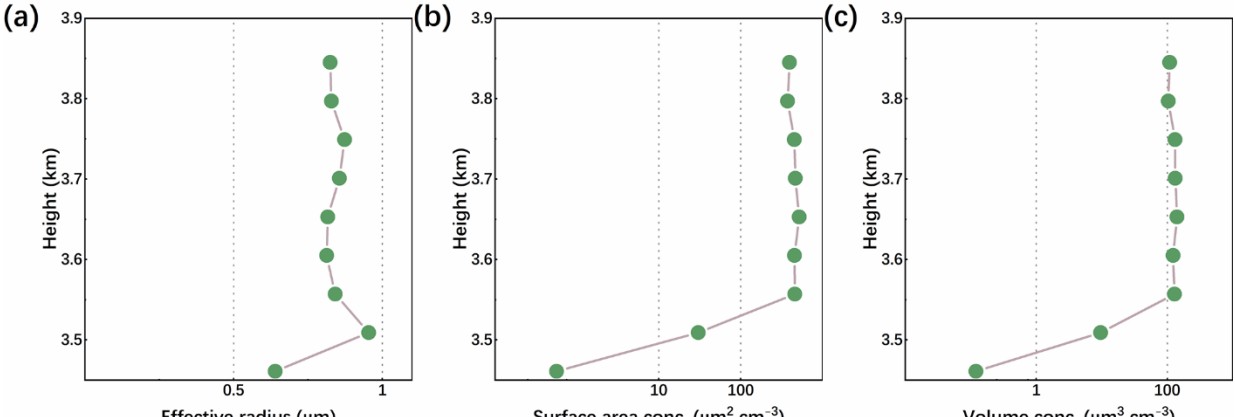

**Figure 9: Marine aerosol AMP inversion from ACDL data products**

### 5.2 Comparison

In order to verify the inversion results of ACDL, the aerosol effective radius results obtained from the inversion are compared with the effective radii of four typical aerosols given by LIVAS and CALIPSO. ACDL presents the inverse result of the profile, reflecting microphysical properties in the vertical structure, whereas both LIVAS and CALIPSO give the mean value. The mean effective radius of the ACDL inversions is thus calculated, along with the standard deviations. It is important to highlight that there are inherent discrepancies between models that assess aerosol particle parameters and the actual spatial and temporal distributions. In this comparison, we are examining the general trends rather than the precise values of the microphysical properties derived for various types of aerosols. The effective radius retrieved from ACDL are 0.47 µm for urban, 0.57 µm for smoke, 0.61µm for dust, and 0.83 µm for marine aerosols. Figure 10 shows that the effective radius of urban aerosols and smoke derived from the ACDL inversion are sight larger than the values given by CALIPSO and LIVAS. This is due to the fact that both types of urban and smoke are characterised by the dominance of fine mode particles in the aerosol volume distributions proposed by Omar and LIVAS. For particle size ranges from 0.05 µm to 0.3 µm, the scattering of aerosols at wavelengths of 532 nm, 1064 nm and 1572 nm is less sensitive to particles, which leads to a reduction in the accuracy of inversion. In addition, for dust aerosols, the inversion results from ACDL are in good agreement with LIVAS. CALIPSO model gives a higher proportion of fine-mode particles in the dust aerosol distribution, resulting in a slightly smaller than the remaining two data sets. With regard to marine aerosols, the radius results obtained from the ACDL inversion are in close agreement with the values provided by CALIPSO and LIVAS, and there is a strong degree of consistency among the three datasets. In addition, results from ground-based system are also considered for comparison. For urban and smoke aerosol, ground-based multiwavelength Raman lidar studies have generally reported smaller effective radius (Pérez-Ramírez et al., 2021; Veselovskii et al., 2015), however, these studies also pointed out that processes such as aging during transport and hygroscopic growth can significantly increase aerosol particle radius, which indicates that our inversion results likely represent more aged and mixed aerosol. Studies with ground-based lidar measurements provide





effective radius for dust in the range of ~0.6-0.9 µm over regions such as Taklamakan Desert and West Africa (Hu et al.,
2020; Veselovskii et al., 2016), and occasionally approaching 1.0 µm, which agree well with our results.

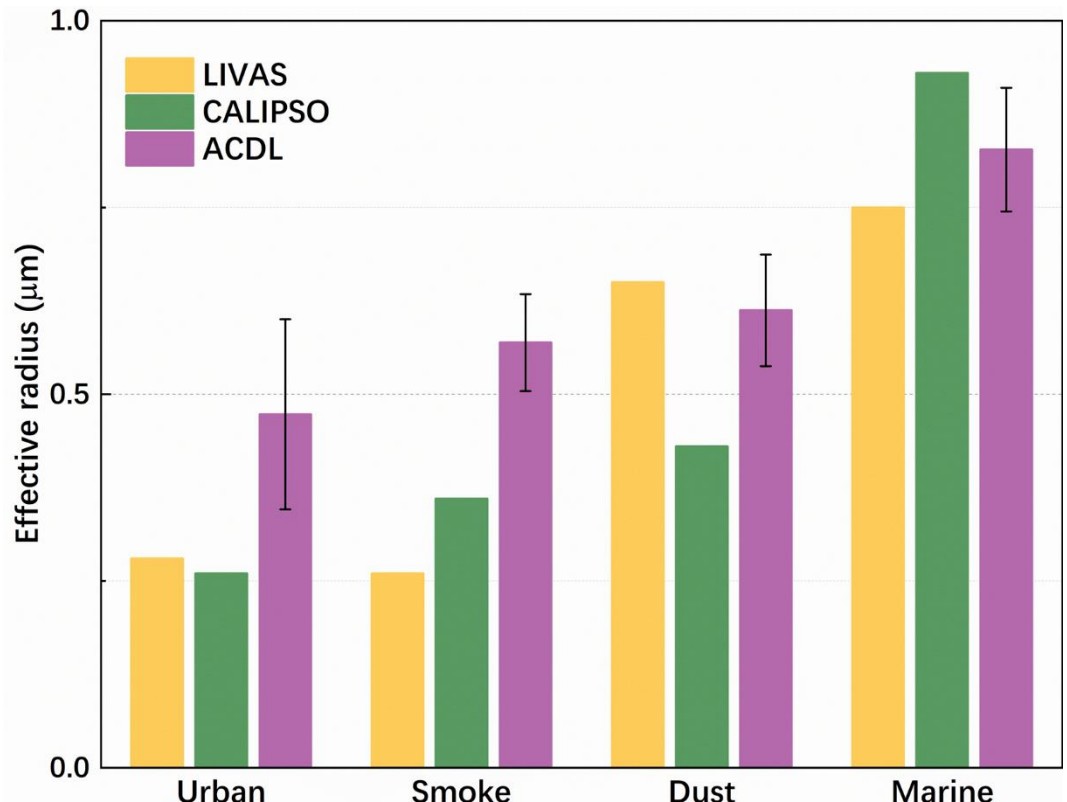

**Figure 10: Comparison of ACDL inversion effective radius results with CALIPSO and LIVAS**

## 6 Summary

In this work, the initial attempt is to derive typical aerosols particle size distribution parameters with vertical resolution from
the first spaceborne HSRL lidar observations based on ACDL onboard AEMS satellite.

A simulation is conducted in advance of the experiment. Unlike the conventional optical data combinations (3β+2α), the
input optical data selected for regularization in this paper are at ACDL operating wavelengths of 532 nm, 1064 nm, and 1572
nm. The simulation demonstrates an improvement in inversion accuracy for coarse mode particle without introducing
supplementary a priori constraints. Inevitably, we lose some information on the fine mode particle. However, the inversion
error of the overall distribution is acceptable. In addition, a sensitivity analysis is carried out, which demonstrates that the
output AMP remained within a reasonable margin of error when the optical data introduced errors in the range of ±20%.

We then carried out case studies of the ACDL. We select scenarios from ACDL's aerosol classification products such as
urban pollution, smoke aerosol intrusion from wildfires into the tropopause, dusty weather, and marine aerosols on the sea





surface, from which we derive and analyse the microphysical properties of the typical aerosols. Furthermore, the credibility

of the inversion results is enhanced through comparison of the effective radius with data from the CALIPSO and LIVAS. The vertical distribution of aerosol particle size parameters in these special weather scenarios is related to the radiative properties of aerosols. Long term continuous observations will facilitate the assessment of the impact of aerosols on climate events and the radiation balance of the Earth-atmosphere.

In future work, we could construct scattering lookup tables for different aerosol shapes based on the depolarization ratio data.

The next stage of research will also focus on improving the quality of inversion for fine mode particles. We look forward to further improving the accuracy of data products for the derivation of aerosol particle size parameters from ACDL.

**Data availability**

Data underlying the results presented in this paper are not publicly available at this time but may be obtained from the authors upon reasonable request.

**Author contribution**

ZB conceived and designed the regularization algorithms; JH prepared the algorithm for inversion and classification of aerosol optical parameters. ZB wrote the manuscript; YX, JL, and WC provided the supervision and participated in the scientific discussion. All the co-authors reviewed and edited the manuscript.

**Competing interests**

The contact author has declared that none of the authors has any competing interests.

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
