# Peer review of "Retrievals of vertically resolved aerosol microphysical particle parameters with regularization from spaceborne Aerosol and Carbon dioxide Detection Lidar (ACDL)"

_EGUsphere, 2025_

## Author Comment (AC1)

**Manuscript number:** egusphere-2025-4208
**MS type:** Research article
**Title:** Retrievals of vertically resolved aerosol microphysical particle parameters with regularization from spaceborne Aerosol and Carbon dioxide Detection Lidar (ACDL)
**Author(s):** Ziyu Bi et al.
**Iteration:** Interactive discussion

We would like to express sincere gratitude to the Reviewers for the careful reading and providing comments. The point-to-point replies to the Referee comments are listed below.

**Replies to Referee comment:**

**RC1.** This inversion technique is widely used for ground based lidars, but it is really a challenge to use it for satellite measurements. The main problem is to provide high quality of input data, because measurements from space are characterized by high noise. So, anybody, who tries to present such inversion, first of all should demonstrate profiles of aerosol backscattering and extinction coefficients with corresponding uncertainties. This is what I miss in this manuscript. Authors should explain how they calculate backscattering and extinction at 1064 and 1572 nm. I could also provide other comments, but question about input data quality is the main. I think, without it manuscript cannot be published.

**Responses:**

Thanks for pointing out this. We fully agree that the quality of the input data is necessary and the method that calculate backscattering and extinction at 1064 and 1572 nm should be explained.

First, we provide an explanation of the retrieval methodology for aerosol extinction and backscattering coefficients at 1064 nm and 1572 nm. The ACDL data processing team follows a structed approach to aerosol inversion tasks. The relevant algorithm has undergone cross-validation which shows good consistency and stability, results for this part are currently under preparation and will be detailed in the subsequent work. Following the suggestion, we have added vertical profiles figure of aerosol backscattering and extinction coefficients, along with their corresponding standard deviation estimates.

The relevant revised version will be displayed in blue font in Section 5.1. Once again, we gratefully thank you for your suggestion. The scientific team of ACDL has also been continuously working on the retrievals of raw data and attempting to incorporate emerging methods to provide high-quality optical data. In addition, as you mentioned, the application of regularization in ground-based lidar has been quite mature, and our work is a preliminary attempt to transfer this to the spaceborne lidar system.

**The revised version:**

In the section 5.1:

"The retrieval methodology for 532 nm channel utilizes the HSRL technique (Dai et al., 2024) and will not be repeated here. For both the 1064 nm and 1572 nm channels, we adopted the Fernald forward integration method. Unlike conventional method, our approach leverages the advantage of the HSRL capability at 532 nm. Specially, the retrieval process first uses the cloud-aerosol classification product retrieved from 532 nm to identify the aerosol type. Then, based on this classification and relevant prior knowledge, an appropriate lidar ratio is selected for the subsequent retrieval. The cross-validation with CALIPSO shows a good consistency and stability as shown in Figure 6. Figure 6 shows the retrieved aerosol extinction coefficients (a) and backscattering coefficients (b). The two sets of profiles are respectively retrieved from ACDL on June 5, 2023 within the latitude of 69.22-70.69, and from CALIPSO on the same day within the latitude of 69.217-70.692. Under the influence of slight temporal and spatial differences, the extinction and backscattering coefficients at the 1064nm obtained from two spaceborne lidar system still have good consistency, and the uncertainty of ACDL is significantly smaller than CALIPSO, which proves the effectiveness of our improved inversion method.

[Figure]

Figure 6: Vertical profiles of aerosol backscattering and extinction coefficients at 1064 nm from ACDL and CALIPSO, along with their corresponding standard deviation estimates.

The vertical profiles aerosol backscattering and extinction coefficients at 532 nm, 1064 nm, 1572 nm obtained from ACDL, along with their corresponding standard deviation estimates, are shown in Figure 7, which selected from ACDL measurements

on March 1st, 2023. Figure 7(a) shows the retrieved aerosol extinction coefficients profile, and Figure 7(b) shows the backscatter coefficients profile. The blue curves represent the 1064 nm signals, and the red curves correspond to 1572 nm. The shaded areas denote the estimated standard deviations. It can be observed that the uncertainty associated with 1572 nm signal is slightly larger than that of the 1064 nm signal. This behavior is physically explainable, the design of the 1572 nm channel is mainly for detecting the column concentration of carbon dioxide, thus, the laser pulse energy at this wavelength is lower compared to which at 532 nm and 1064 nm, leading to the smaller signal-noise ratio. Meanwhile, the reliability of ACDL data has been validated in existing studies (Liu et al., 2024), the deviations of extinction and backscattering coefficients at 532 nm are controlled within approximately 30% and 25% respectively, compared to ground-based lidar. Considering the results of this study and the comparison with CALIPSO in Figure 6, the uncertainty levels presented in Fig.7 are acceptable. These results demonstrate good observational consistency and low systematic errors with ACDL, which provides a solid foundation for the regularization inversion in this work."

[Figure]

Figure 7: Vertical profiles aerosol backscattering and extinction coefficients at 532 nm, 1064 nm, 1572 nm obtained from ACDL, along with their corresponding standard deviation estimates.

Dai, G., Wu, S., Long, W., Liu, J., Xie, Y., Sun, K., Meng, F., Song, X., Huang, Z., and Chen, W.: Aerosol and cloud data processing and optical property retrieval algorithms for the spaceborne ACDL/DQ-1, Atmospheric Measurement Techniques, 17, 1879-

1890, 10.5194/amt-17-1879-2024, 2024.

Liu, Q., Huang, Z., Liu, J., Chen, W., Dong, Q., Wu, S., Dai, G., Li, M., Li, W., Li, Z., Song, X., and Xie, Y.: Validation of initial observation from the first spaceborne high-spectral-resolution lidar with a ground-based lidar network, Atmos. Meas. Tech., 17, 1403-1417, doi:10.5194/amt-17-1403-2024, 2024.

---

## Author Comment (AC2)

Manuscript number: egusphere-2025-4208

MS type: Research article

**Title:** Retrievals of vertically resolved aerosol microphysical particle parameters with regularization from spaceborne Aerosol and Carbon dioxide Detection Lidar (ACDL)

**Author(s):** Ziyu Bi et al. **Iteration:** Final response

We would like to express sincere gratitude to the Reviewers for the careful reading and providing comments. The point-to-point replies to the Referee comments are listed below.

**Replies to RC2:**

The manuscript represents the application of the regularization algorithm to the first spaceborne HSRL lidar ACDL, and enabling the retrievals of vertically resolved aerosol microphysical parameters. Although there is potential to improve the retrieval of the fine-mode particles, this work shows a valuable contribution by obtaining AMP from the spaceborne lidar and it is interesting. Before the manuscript can be publication, the following questions need to be addressed.

1. The author's choice of wavelengths in the inversion process differs from the commonly scheme  $(3\beta+2\alpha)$ . Please explain the basis for the selection of the input data combination in this paper.

**Responses:**

Thanks for the comment regarding the wavelength selection in the inversion. The choice of wavelength combination is not arbitrary but is considered by the ACDL instrument architecture and to provide better constraint on coarse mode retrievals.

On an instrument base, in most previous studies, the wavelength combination at 355 nm, 1064 nm, and 1064 nm was employed because these wavelengths are available on ground-based systems. However, the ACDL onboard the AEMS satellite operates with laser wavelength at 532 nm, 1064 nm, and 1572 nm. On a physical base, the extinction and backscatter efficiencies at different wavelengths are sensitive to the particles with different size. Previous studies show that additional measurement channels help extend the retrieval size range (Veselovskii et al., 2004), and adding longer near-infrared wavelength can improve the coarse-mode AMP (Böckmann et al., 2024). Our simulation results confirm this behavior as shown in Fig. 2. Therefore, the optical data combination adopted directly follows the instrument design and all available wavelengths are considered in this work.

**2.** In the actual atmosphere, the refractive index of the atmosphere varies significantly due to changes of humidity and the type of aerosols, and there is a lack of discussion on the sensitivity of the inversion results to the choice of refractive index.

**Responses:**

We appreciate the reviewer concern. The aerosol complex refractive index m does vary greatly with its composition. And the sensitivity of inversion to the assumed m has been investigated in previous studies, which demonstrated that the uncertainties in m assumptions can be reduced by expanded wavelength combinations and other optimized constraints (Pérez-Ramírez et al., 2013; Pérez-Ramírez et al., 2020; Whiteman et al., 2018). In addition, we consider that uncertainties in m assumptions is not the main point we concerned, because in this work, we do not assume m to a single value for inversion, but adopt all possible m to retrieve distributions and then selected the acceptable solutions by the method described as Eq. (6) in Sec. 3. This approach is proposed by Müller et al. (Müller et al., 1999) and Veselovskii et al. (Veselovskii et al., 2002)and has been widely used (Di et al., 2018; Yan et al., 2019). And for the logical flow of the manuscript, we have added the relevant explain in Sec. 3 (lines 164) and the content is highlighted in blue font.

**The revised version:**

In Section 3 (lines 168-178):

"While the complex refractive index in the actual atmosphere is difficult to obtain, and the stand-alone lidar inversion is sensitive to the assumed complex refractive index. Thus, this work does not assume complex refractive index to a single value for inversion, but adopt all possible complex refractive index (for real parts of the refractive index, values of 1.3-1.6 are used, and for the imaginary part, values of 0.001-0.2 are chosen) to retrieve distributions and then selected the acceptable solutions by the method described as Eq. (6). This procedure has been widely used in previous studies (Di et al., 2018; Yan et al., 2019)."

**3.** Please provide a further explanation for the increase in the bimodal inversion errors in the simulation (volume concentration reaching 60%).

**Responses:**

We thank the reviewer for this valuable comment. The reason for the increase of the inversion errors in the bimodal can be summarized as the challenge in bimodal retrieval and the algorithm limitation.

Numerous studies have reached a consensus on the difficulty of the bimodal distribution retrieval and it's a challenge in the field. The inversion errors of the bimodal distribution are generally larger than the unimodal. Because with two modes, the kernel function for backscatter and extinction at a few wavelengths are overlapped in the large radius range, and leads to more degrees of freedom without any other constraints in the

retrieval (Müller et al., 1999; Veselovskii et al., 2002). Besides, the solution averaging process performed in the algorithm tends to influence the peak and the mode width of the reconstructed distribution (Veselovskii et al., 2004). And volume concentration scales with  $r^3$  as a size-integrated parameters. Thus, the small reconstructed distribution errors will be accumulated, and the inversion error for the volume concentration will be larger than effective radius and the surface area concentration. This behavior has consistency with the results reported by Di et al. and Yan et al. Although the errors of bimodal volume concentration reach near the 60%, the 92% of which are controlled below 40% (as shown in Fig. 2). This error range is reasonable and acceptable compared with previous studies. We have added more discussion in Sec. 4.2.

**The revised version:**

In Section 4.2 (lines 240-258) and highlighted in blue font:

"For the bimodal distribution, variation of the scattering properties become more complicated and the inversion becomes more ill-posed. It is difficult to perform stable retrieval with a few kernels function (equal to the number of the input optical data at different wavelengths and the base function) due to insufficient optical constraints, because they overlap in radius range, which makes separate the fine and coarse mode hard. Regularization stabilizes the solution but tends to smooth the peaks and influence the mode width of the reconstructed bimodal distribution near the retrieval radius edge (as shown in Fig. 1). Thus, the reconstruction results of the bimodal distribution show more differences from the original APSD, and leads to an increased inversion errors for AMP calculated based on Eq. (8)-(10). Besides, for both the unimodal and the bimodal distribution, the errors for volume concentration are slight larger than the errors for effective radius as shown in Fig. 2, among the AMP, volume concentration is a direct integral of APSD (Eq. (10)) and the effective radius is a ratio of integrals (Eq. (8)). Thus, the small reconstructed distribution errors will be accumulated, and the inversion error for the volume concentration will be larger than effective radius. This behavior has consistency with the results reported by Di et al (Di et al., 2018)."

**4.** There is a lack of quantitative description of the ACDL data errors shown in Fig. 10, as well as the other two sets of comparison data.

**Responses:**

Thanks for pointing out this shortcoming. We have added a relevant text in Sec. 5.2 to provide a quantitative and clear description.

**The revised version:**

In Section 5.2 (lines 401-412) and highlighted in blue font:

"In order to verify the inversion results of ACDL, the aerosol effective radius results obtained from the inversion are compared with the effective radius of four

typical aerosols given by LIVAS (Amiridis et al., 2015) and CALIPSO (Omar et al., 2009; Omar et al., 2005). And the comparison results are shown in Fig. 10. For urban aerosol, effective radius is 0.28  $\mu$ m from LIVAS and 0.26  $\mu$ m from CALIPSO, while retrieved results from ACDL is 0.47±0.127  $\mu$ m. For smoke aerosol, the effective radius results are 0.26  $\mu$ m, 0.36  $\mu$ m, and 0.57±0.065  $\mu$ m from LIVAS, CALIPSO, and ACDL, respectively. The effective radius for dust is 0.65  $\mu$ m from LIVAS, 0.36  $\mu$ m from CALIPSO and 0.61±0.075  $\mu$ m from ACDL. The results for marine aerosol are 0.75  $\mu$ m, 0.93  $\mu$ m, and 0.83±0.083  $\mu$ m, respectively."

**5.** It would be better to add the display of parameters for the ACDL system. **Responses:**

Thanks for the valuable suggestion. We have added a table of ACDL parameters in Sec. 2.1 to provide more information for the instrument.

**The revised version:**

In Sec. 2.1 (line 96) and highlighted in blue font:

"Table 1 shows the main parameters of the ACDL system."

Table 1. Main parameters of the ACDL system

[revised manuscript text omitted]

---

## Author Response (AR2)

**Manuscript number:** egusphere-2025-4208
**MS type:** Research article
**Title:** Retrievals of vertically resolved aerosol microphysical particle parameters with regularization from spaceborne Aerosol and Carbon dioxide Detection Lidar (ACDL)
**Author(s):** Ziyu Bi et al.
**Iteration:** Minor revision

Dear Editor:

We sincerely thank the editor and all reviewers for their valuable feedback that we have used to improve the quality of our manuscript. All comments have been addressed below, and modifications have been made accordingly. The point-by-point replies are listed below.

Thanks very much for considering this work! Kind regards!

Ziyu Bi

On behalf of the co-authors

**Referee #3:**

First, I apologize for my late answer. I have read the paper carefully and most of my previous concerns have been addressed. Now I believe the paper is suitable for publication. There is just one minor issue I would like the authors to try to address. In many previous studies of aerosol microphysical properties retrievals, it was used 355 nm. For the ACDL this wavelength does not operate, but it includes 1572 nm. That is fine. But I wonder if the lack of measurements in 355 nm could force the retrieval to underestimate the fine mode. Could this be the reason behind the large discrepancies in effective radius for urban and biomass-burning aerosols?

**Responses:**

We thank the reviewer for this valuable question. We agree that absence of the 355 nm may force the retrieval to underestimate the fine mode.

Indeed, 355 nm has been widely used for its strong sensitivity to submicron aerosol particles in the fine mode, and previous studies have shown that when the wavelength combination lacks sufficient spectral coverage (or the input information is limited) the inversion results trend to exhibit large errors (Veselovskii et al., 2004), and the combination included 355 nm extinction display more sensitivity to smaller particles than which without it in retrievals (Whiteman et al., 2018).

As the reviewer points out, the ACDL system does not include 355 nm and the lack of the 355 nm channel inevitably reduces sensitivity to the lower end of the fine-mode spectrum, where aerosol extinction and backscatter efficiency is significantly higher in the UV (the peak values of extinction and backscatter efficiency at 355 nm are ~0.48 μm, and ~0.92 μm, respectively) (Di et al., 2018). And when the fine mode

particles are dominant, as is typically the case for urban and biomass-burning aerosols, our wavelength combination for input optical parameters may underestimate the relative contribution of small particles and shift the reconstructed fine mode slightly toward larger radius range, which in return elevates the effective radius.

To address this issue, we have added the relevant discussion in Sec. 5.2 (lines 404) and the content is highlighted in blue font.

**The revised version:**

In Section 5.2 (lines 404-412):

"This is because both urban and smoke aerosols are characterized by the dominance of fine-mode particles, as indicated by the aerosol volume distributions from Omar et al. and the LIVAS climatology. In this study, the retrieval is performed based on a wavelength combination of 532 nm, 1064 nm, and 1572 nm. Compared with conventional combinations included 355 nm, which exhibits stronger scattering sensitivity to smaller aerosol particles for retrievals, with t extinction and backscatter efficiency peaks at ~0.48 μm, and ~0.92 μm, respectively (Di et al., 2018; Whiteman et al., 2018), our configuration provides reduced sensitivity to fine mode particles. As a result, the absence of the 355 nm channel may lead to an underestimation of the relative contribution of fine-mode particles and a slight shift of the reconstructed distribution toward larger radius, resulting larger discrepancies in effective radius for urban and smoke aerosols."

Di, H., Wang, Q., Hua, H., Li, S., Yan, Q., Liu, J., Song, Y., and Hua, D.: Aerosol Microphysical Particle Parameter Inversion and Error Analysis Based on Remote Sensing Data, Remote Sensing, 10, 10.3390/rs10111753, 2018.

Veselovskii, I., Kolgotin, A., Griaznov, V., Müller, D., Franke, K., and Whiteman, D. N.: Inversion of multiwavelength Raman lidar data for retrieval of bimodal aerosol size distribution, Appl. Opt., 43, 1180-1195, doi:10.1364/AO.43.001180, 2004.

Whiteman, D. N., Pérez-Ramírez, D., Veselovskii, I., Colarco, P., and Buchard, V.: Retrievals of aerosol microphysics from simulations of spaceborne multiwavelength lidar measurements, Journal of Quantitative Spectroscopy and Radiative Transfer, 205, 27-39, 10.1016/j.jqsrt.2017.09.009, 2018.